# A Novel Feature-Engineered–NGBoost Machine-Learning Framework for Fraud Detection in Electric Power Consumption Data

**DOI:** 10.3390/s21248423

**Published:** 2021-12-17

**Authors:** Saddam Hussain, Mohd Wazir Mustafa, Khalil Hamdi Ateyeh Al-Shqeerat, Faisal Saeed, Bander Ali Saleh Al-rimy

**Affiliations:** 1School of Electrical Engineering, University Technology Malaysia, Johor Bahru 81310, Malaysia; wazir@utm.my; 2Department of Computer Science, College of Computer, Qassim University, Buraydah 51452, Saudi Arabia; kh.alshqeerat@qu.edu.sa; 3School of Computing and Digital Technology, Birmingham City University, Birmingham B4 7XG, UK; alsamet.faisal@gmail.com; 4School of Computing, Faculty of Engineering, Universiti Teknologi Malaysia, Johor Bahru 81310, Malaysia

**Keywords:** theft detection in power consumption data, NGBoost algorithm, majority weighted minority oversampling technique algorithm, whale optimization algorithm, tree SHAP algorithm

## Abstract

This study presents a novel feature-engineered–natural gradient descent ensemble-boosting (NGBoost) machine-learning framework for detecting fraud in power consumption data. The proposed framework was sequentially executed in three stages: data pre-processing, feature engineering, and model evaluation. It utilized the random forest algorithm-based imputation technique initially to impute the missing data entries in the acquired smart meter dataset. In the second phase, the majority weighted minority oversampling technique (MWMOTE) algorithm was used to avoid an unequal distribution of data samples among different classes. The time-series feature-extraction library and whale optimization algorithm were utilized to extract and select the most relevant features from the kWh reading of consumers. Once the most relevant features were acquired, the model training and testing process was initiated by using the NGBoost algorithm to classify the consumers into two distinct categories (“Healthy” and “Theft”). Finally, each input feature’s impact (positive or negative) in predicting the target variable was recognized with the tree SHAP additive-explanations algorithm. The proposed framework achieved an accuracy of 93%, recall of 91%, and precision of 95%, which was greater than all the competing models, and thus validated its efficacy and significance in the studied field of research.

## 1. Introduction

The quality of living in modern society is highly associated with the availability of electricity [1]. An uninterrupted electricity supply requires an efficient transmission and distribution (T&D) infrastructure. Broadly, there are two types of losses in any T&D system; i.e., technical and nontechnical losses. The technical losses account for the heating effect in the resistive nature of T&D lines, transformers, and other equipment [2]. On the other hand, nontechnical losses (NTLs) occur due to equipment installation errors, billing irregularities, corruption within company staff, and electric theft. Among all the mentioned causes of NTLs, the theft of electric power is the most severe issue faced by power utilities around the globe. As an estimation, electric power theft causes an annual loss of more than USD fifty billion worldwide [3,4]. Figure 1 illustrates the severity of this problem in various countries.

Researchers have explored several different methods in the literature to overcome the stated issue, among which the hardware and data-driven approaches are the most widely utilized theft-detection strategies [5]. Hardware-based solutions employ numerous instruments such as balancing or calibration meters, sensors, tamper-evident lock detectors [6], and other devices to detect electric fraud [7]. In these methods, the mentioned devices assist in aggregating the various network characteristic values at different locations so that an immediate discrepancy report can be generated during the occurrence of theft. Even though these methods have reported suitable outcomes, they are not feasible for underdeveloped countries due to the high initial and maintenance costs. Contrary to hardware-based approaches, the data-driven theft-detection techniques only require the accumulated kWh data of consumers to perform the same task. The data-driven approaches can be broadly categorized into unsupervised and supervised machine-learning (ML) methods. The former methods utilize the unlabeled dataset acquired from energy meters to classify the theft and healthy consumers by employing various similarity or dissimilarity metrics through a clustering approach [8]. On the other hand, the supervised ML methods utilize a prelabeled dataset of consumers’ consumption profiles to train the classifier initially, which, at a later stage, assists in identifying the suspicious consumption profiles in the provided dataset.

It is worth noting that, in comparison to unsupervised machine-learning-based fraud-detection models, supervised machine-learning approaches are the most widely used, owing to their flexibility of training a classifier on multiple classes, ease of execution, and higher interpretability of model outcomes. However, the supervised machine-learning-based electric-theft-detection frameworks encounter few challenges during each of their execution stages; i.e., data preprocessing, feature engineering, and model training and testing. For example, the data acquired from the conventional or smart meters generally contain a lot of missing and inconsistent observations that need to be imputed for achieving realistic outcomes at the end of the classification process. Imbalanced data class distribution and inappropriate selection of the features may also prevent the effective and unbiased evaluation of the theft-detection model. Another important aspect that has been ignored in most of the supervised ML theft-detection frameworks is the probability evaluation of model outcomes. The greater the model outcome probability, the greater the sureness on the produced model prediction, leading to a reduced number of false inspections. Moreover, the contribution of input features on a particular predicted outcome is often overlooked, which consequently impedes the further improvement of the developed model.

### Contributions of the Proposed Theft-Detection System

The broader aim of this research work was the implementation of a novel sequentially executed theft-detection framework to facilitate the power utilities in their campaign against fraudster consumers. The proposed framework initially utilized an ML-based random forest imputer (RFI) to impute the missing entries in the acquired smart meter dataset (SGCC dataset). The RFI is an effective technique to handle different types of missing values by comprehending complex interactions present in the data. To avoid the data class unbalancing, the majority weighted minority oversampling technique (MWMOTE) algorithm was utilized. The MWMOTE algorithm employs the intelligence of the average-linkage, agglomerative-clustering-based technique to generate the required number of minority class samples from the original data to effectively balance the overall data class distribution. In order to precisely portray the underlying characteristics present in consumption data, the proposed approach utilized the intelligence of the time-series feature-extraction library (TSFEL) for extracting the statistical, temporal, and spectral domain-based features from users’ kWh consumption patterns. The whale optimization algorithm (WOA)-based feature-selection method was adopted to avoid overfitting and high data dimensionality by selecting the most significant features that positively contributed towards predicting the target variables, while discarding the less relevant ones. Once the most relevant features were acquired, the model training and testing process was initiated by using the NGBoost algorithm to classify the consumers into two distinct categories (“Healthy” and “Theft”). Finally, unlike most of the ensemble and gradient-boosting-based ML models in which the predictions are made as a black box; i.e., the reason for any given prediction was not thoroughly explained, this study utilized the tree Shapley additive explanations (SHAP) algorithm to evaluate the impact of each input feature in predicting the target variable. The proposed model achieves an excellent accuracy, with low false-positive and high detection rates; thus, it saves the cost, labor, effort, and time required for executing onsite inspections.

The remainder of the paper is organized into five main sections. Section 2 provides the basics and the recent advancements in the studied research area. Section 3 explains the research methodology. In Section 4, the performance evaluation of the proposed theft-detection framework is provided. Section 5 illustrates the results and a discussion, while conclusions of the current research work are laid in Section 6.

## 2. Literature Review

As this study explores an application of the supervised ML approach in detecting theft cases from the acquired smart meter dataset, therefore the literature pertaining to the recent advancements in the mentioned field is discussed in detail in this section. As mentioned in the previous section, the supervised machine-learning-based electric-theft-detection frameworks encounter a few major challenges during data preprocessing, feature engineering, and the model training and testing process. It is important to discuss each of the challenges and their available solutions in the literature in detail to highlight the essence of current research work.

To achieve realistic outcomes at the end of any classification process, the missing observations in the accumulated dataset need to be logically and intelligently imputed. Since most of the publicly available electric consumption datasets possess a considerable chunk of missing data entries, it therefore is very hard for a classifier to learn and classify such deficient and inconsistent datasets. In order to overcome the stated issue, several data-imputation or data-dropping approaches have been suggested in the literature. These approaches include the look-back and sandwich-based imputation methods [9], Monte Carlo technique [10], fuzzy clustering method [11], ensemble of multilayer perceptron [12], Bayesian missing values estimation [13], hot deck [14], and mean imputation. Even though the stated solutions are easy to implement and are computationally fast, they cause a substantial data loss, decreased statistical power, increased standard errors, and reduced model generalizing ability, thereby yielding misleading conclusions [15].

The second stage in any supervised ML method is the feature-engineering stage. In this stage, several distinct features are extracted from the acquired dataset using different soft computational techniques. The major challenge during this stage is to deal with the imbalanced data class distribution. In addition, the selection of a suitable technique for choosing the most relevant features from the provided dataset is highly important. The former issue occurred due to the higher number of healthy consumers than fraudster consumers in the acquired dataset, which made the model highly biased towards the majority-class sample, creating a low representation of the minority-class samples. Since the under-represented class in any theft detection framework needs to be identified, a balanced data class distribution is essentially required for the effective and unbiased evaluation of any theft-detection model. Several researchers have attempted to tackle this challenge using different soft computational methods. Glauner et al. [16] addressed the data class imbalance issue by training the various ML classifiers on the different proportions of NTL cases. The authors chose the area under the receiver operating characteristic curve (AUC) metric to assess the performance of the studied classifiers in the presence of a data class imbalance. Hasan et al. [17] and Gunturi et al. [18] employed the over-sampling-based technique using the synthetic minority oversampling technique (SMOTE) algorithm to tackle the mentioned issue. In another study, Buzau et al. [19] used the undersampling-based technique to balance the considered dataset. However, the oversampling-based class balancing techniques generally cause overfitting, low generalization ability, and noisy data generation. In contrast, the undersampling-based class balancing techniques cause a substantial loss of information, consequently lowering the developed model’s accuracy [20].

Another challenge that emerges during the feature-engineering stage is the choice of an appropriate method for extracting and selecting the most relevant features from the acquired dataset. The raw data obtained from the smart meters generally lacked statistical significance, and contained high data dimensionality with redundant and irrelevant features. If such a dataset is directly provided to the classifier, its performance in classifying the healthy and theft patterns will be highly affected. Therefore, a number of highly relevant features are extracted from the acquired raw dataset to enhance the classifier’s performance. In the literature, several statistical and deep-learning-based techniques have been explored for extracting the supplementary information from the given data [17,21]. However, these statistical-based feature-extraction techniques can only extract simple features such as mean, mode, median, interquartile range, etc. The mentioned features contain less significant information, and are not enough to provide accurate and precise data to the classifier. The authors explored several deep-learning-based techniques to obtain some significant features that provided in-depth data insights to the classifier to overcome this issue. Nevertheless, the implementation of these techniques was too complicated and computationally expensive to pursue.

On completion of the feature-engineering procedure, the next challenge in developing an efficient theft-detection framework was to shortlist the fraudster consumers by using a suitable ML classifier. Numerous ML classifiers were utilized in the literature for developing an effective theft-detection framework. Jindal et al. [22] and Marimuthu et al. [23] presented an energy-theft-detection model using a support vector machine (SVM) for the smart meter dataset. Salman et al. [24] and Yan et al. [25] utilized ensemble machine-learning-based techniques employing random forest and Xgboost algorithms to enhance the classification performance of developed theft-detection frameworks. In [26], the authors developed a theft-detection framework using the C5.0 boosting algorithm, and indicated the performance enhancement before and after using feature-engineering techniques. Pereira et al. [27] utilized a PSO-based hyperparameter tuned MLP network to identify theft cases in the Brazilian distribution network and achieved an accuracy of 94.58%. Similarly, Jokar et al. [28] proposed a consumption-pattern-based energy-theft detector (CPBETD) algorithm to detect fraudster consumers using the Irish smart meter dataset, and obtained a recall value of 94%. In the mentioned research works above, the developed models were designed in such a way so as to yield the single best-guess prediction or point estimate; e.g., “Healthy” or “Theft”, thus providing a factor of uncertainty in the predicted outcomes. Furthermore, such classification procedures may cause a substantial number of false inspections, leading to increased expenses, wastage of time, and lack of confidence in the developed model. To overcome the stated issues, probabilistic theft-detection models are often used. The probabilistic models are more reasonable models in terms of gauging the model’s confidence in the predicted outcome. This is because they assist in generating a complete probability distribution function over the entire classifier outcome. For most of the ensemble and gradient-boosting-based ML models, the predictions are treated as black boxes; i.e., the reason for any given prediction is unknown; thus, the chance for further improvement is nullified.

Concluding the detailed literature review, the mapping of the identified problems and their proposed novel solutions are presented in a tabular form in Table 1, to highlight the essence of the current research work.

## 3. Proposed Methodology

The overall framework was classified into three major stages: data preprocessing, feature engineering, and model evaluation, as shown in Figure 2. This section discusses all stages of the designed model in detail.

### 3.1. Stage-1: Data Preprocessing

In data preprocessing, the original data is reshaped into an appropriate representation useful for the effective learning of ML models. In the current research work, the consumers’ kWh consumption data (January 2014 to October 2016), acquired from the State Grid Corporation of China (SGCC) [40], contained 42,372 labeled consumers (91% healthy and 9% theft). In order to explore the acquired labeled data, the consumption patterns for a few of the random samples from both fraudster and healthy consumers were plotted, as shown in Figure 3 and Figure 4, respectively.

One of the key characteristics of theft consumers that most likely distinguished them from the healthy ones was their unsymmetrical energy consumption pattern. It can be observed in the mentioned figures that the considered fraudster samples contained comparatively more nonperiodicity and zero consumption values than the healthy consumers, thus validating the genuineness of the considered dataset. Once the labeled dataset was explored, the next task for developing an efficient supervised ML classification framework was to accurately impute the missing entries in the acquired data. Figure 5 shows the histogram for missing values present in the accumulated dataset.

It can be observed in Figure 5 that each amongst the 60.11% of the total consumers contained less than 200 missing data (NaN) entries, while this number ranged between 300 and 600 and 695 and 705 for 16.89% and 23% of consumers, respectively. Since it was quite difficult to accurately impute such a huge number of missing data entries, only those consumers whose NaN entries were less than 200 were shortlisted for further processing. The current study utilized the random forest imputation (RFI) technique [36] to impute the missing entries in kWh consumption data for these selected consumers. The mentioned task was accomplished by using the Miss-Forest package present in the R programming language. It created a data matrix, $\left[n\times p\right]$, where *n* is the number of consumers and p is the number of columns in kWh consumption for each consumer. Initially, the missing values present in the kWh consumption data were replaced by either the mean or median value. Afterward, the imputation process was initiated sequentially for each imputed variable in such a way that the variable under imputation was used as the target variable for building the RF model on the remaining variables. Subsequently, the variable value estimated by the trained RF model was replaced with the imputed value. This process was repeated until all the predefined number of iterations were completed. Figure 6 depicts the kWh consumption patterns of four random consumers before and after the proposed imputation process.

### 3.2. Stage-2: Data Class Balance and Feature Engineering

This stage was further divided into two substages; i.e., data class balancing and feature engineering. This section discusses each stage in detail.

#### 3.2.1. Data Class Balancing

Data class balancing is the process of balancing the minority and majority class samples with each other to improve the generalization ability of the model and avoid overfitting issues. In this study, the acquired prelabeled data contained only 6.7% of the theft consumers, which was comparatively lower than the healthy consumers. To increase the minority class samples, the oversampling technique based on the MWMOTE algorithm was utilized. The working mechanism of the MWMOTE algorithm is based on three stages.

In the first stage, the identification process of highly important minority class samples was carried out. In the second stage, samples were categorized in terms of their Euclidean distance from the nearest majority class sample. Finally, the average-linkage agglomerative clustering-based technique was employed to produce new samples from highly important categories of minority class samples formed in the second stage. The minority class samples; i.e., synthetic theft cases generated at the end of this process, were merged with the original dataset to balance the overall data class distribution. The data class distribution before and after using the MWMOTE algorithm was visualized using the t-distributed stochastic neighbor embedding (t-SNE) approach, as shown in Figure 7 and Figure 8.

Figure 7 shows the imbalanced smart meter dataset obtained from SGCC, while Figure 8 depicts a balanced dataset obtained after the implementation of the MWMOTE algorithm. As can be seen from the mentioned figures, the MWMOTE algorithm systematically imputed the minority class samples to achieve a balanced dataset that could be effectively utilized for training and testing the proposed classifier.

#### 3.2.2. Proposed Feature-Engineering Method

In this subsection, the proposed feature engineering procedure is discussed in detail. Feature engineering is the process of selecting the most relevant features and discarding the redundant and less significant ones from the acquired dataset. This procedure aims to boost the learning ability of the ML model while dealing with complex data patterns. The SGCC dataset utilized in this study lacked statistical characteristics. For achieving an efficient performance by any theft-detection model, its input features must reflect sufficient underlying abnormalities in customer consumption data. Therefore, in this study, statistical, temporal, and several spectral domain-based features (more than 50) were extracted from each consumer’s consumption data using the TSFEL technique. The TSFEL technique facilitated rapid data exploration and automatic feature extraction from the given kWh consumption data. All the extracted features in the current study are depicted in Figure 9, whereas the source code and detailed description of each computed feature can be found on the TSFEL GitHub web page [33].

The newly added supplementary information obtained after employing the TSFEL technique enhanced the learning ability of the model in classifying the complex data patterns effectively. Nevertheless, increasing the number of features increased the data dimensions, training time, and computational resources. To overcome this challenge, a feature selection approach was adopted at the later stage to select a small subset of the extracted features. The resulting shortlisted features were generally packed with high-quality and the most relevant information for data class prediction. These feature selection techniques have been found to be effective in avoiding overfitting of the model, lowering the computational and storage requirements, mitigating problems caused due to high data dimensions, and achieving improved readability and interpretability of the model.

In this study, the most essential features from the available kWh consumption data were selected by using the WOA-based feature selection (FS) technique. The WOA-FS technique worked iteratively to select the essential features from the given dataset. The WOA is a stochastic population-based metaheuristic optimization algorithm whose working mechanism is based on mimicking humpback whales’ prey-hunting behavior. It consists of two stages: exploration (random hunting for prey) and exploitation (encircling and attacking prey). Like other population-based metaheuristic algorithms, it iteratively generates random solutions within the bounded search space until the optimum solution is achieved. A simplified working mechanism of the proposed WOA-FS technique is illustrated in Figure 10.

The proposed WOA-FS optimally selected those minimum number of highly important features that attained maximum classification accuracy for any given volume of consumption data and extracted features (as depicted in the square matrix, where Con_n_f_n_ represents Con_n_: consumer number and f_n_: feature number). Before initiating the feature selection process through WOA, the number of iterations, number of search agents, convergence criteria, search space boundaries, decision variables, and the learning classifier had to be decided. During the first iteration, the random subsets of features were selected from consumers’ consumption data and their extracted features to train and test the learning classifier. Afterward, the performance of the classifier with each feature subset was evaluated using the fitness function (FF) provided in Equation (1):
(1)$$\mathrm{FF}=\alpha \times {\mathrm{CLF}}_{e}+\beta \frac{C}{N}$$
where CLF_e_ represents the classification error rate of the classifier (NGBoost in the current case), C represents the cardinality of the chosen feature subset, and N denotes the total input features. The α and β (1−α) manage the trade-off between the classification error rate to the number of selected features subset [41].

A number of solutions [42,43] have been suggested to prevent premature convergence in optimization-based algorithms, with one particularly significant study being carried out by Zhang et al. [44], who proposed that in the case of a premature convergence problem, the best existing solution should be preserved and the mutation process should continue until an improved solution is found; once an improved solution is found, the current optimal solution is updated, and the mutation process is stopped. In the current study, in order to avoid the problem of premature convergence, humpback whales in the WOA technique, during the prey search phase, searched for prey in a random manner according to their relative positions to one another. The WOA forced the search agent to move away from a reference whale by using random values higher than or lower than 1. As a consequence, rather than selecting the best search agent found so far, the WOA changed a search agent’s position in the exploration phase to that of a randomly selected search agent. This WOA’s capacity to randomly explore the solution space, even when it was close to the optimal solution, allowed it to retain population diversity and avoid the premature convergence issue.

It is worth mentioning here that the minimization of the classification error with the limited number of essential features was taken as the current optimization objective. The WOA accomplished the mentioned task by evaluating the FF’s magnitude for each iteration search agent. The search agent with a minimum value of FF was taken as the best solution candidate. Similarly, for each subsequent iteration, the process was repeated until the predecided termination criterion was met. Finally, the selection of the feature subset was made by choosing the fittest solution obtained at the end of the optimization process. The final feature set obtained at the end of the feature selection process contained several data points that were spread across a broad spectrum. Such features with higher magnitude could induce bias during the model training. Therefore, all the computed features had to be standardized on a common uniform scale. The current study employed the well-known min–max approach for data standardization to address this challenge, using the following expression:
(2)$$f\left(x_{i}\right)=\frac{d_{i}-\min\left(D\right)}{\max\left(D\right)-\min\left(D\right)}$$
where D is a vector composed of d_i_ daily electricity consumption, while the min(D) and max(D) are the minimum and maximum values of D, respectively.

### 3.3. Stage-3: Model Training and Evaluation Stage

The theoretical background of the performance metrics and proposed NTL detection classifier are discussed in detail in the following sections.

#### 3.3.1. Performance Evaluation Metrics

The efficacy of the supervised machine learning models depended on their ability to predict the unlabeled data. Numerous metrics exist in the literature for evaluating the performance of an ML model, such as those utilized in [45]. Since it was not feasible to consider all the performance evaluation metrics provided in the cited reference, only a small number of the most significant metrics were employed to assess and compare the efficacy of the proposed theft-detection model. Equations (3)–(11) provide the mathematical expressions for calculating the stated metrics:
(3)$$\mathrm{Accuracy}=\frac{\mathrm{TP}+\mathrm{TN}}{\mathrm{TP}+\mathrm{TN}+\mathrm{FP}+\mathrm{FN}}$$
(4)$$\text{Recall or Detection rate}=\frac{\mathrm{TP}}{\mathrm{TP}+\mathrm{FN}}$$
(5)$$\text{False positive rate}=\mathrm{FPR}=\frac{\mathrm{FP}}{\mathrm{FP}+\mathrm{TN}}$$
(6)$$\text{False negative rate}=\mathrm{FNR}=\frac{\mathrm{FN}}{\mathrm{FN}+\mathrm{TP}}$$
(7)$$\mathrm{Precision}=\mathrm{PR}=\frac{\mathrm{TP}}{\mathrm{TP}+\mathrm{FP}}$$
(8)$$F1_{\mathrm{score}}=2\times \frac{\mathrm{Precision}\times \mathrm{DR}}{\mathrm{Precision}+\mathrm{DR}}=\frac{2\mathrm{TP}}{2\mathrm{TP}+\mathrm{FP}+\mathrm{FN}}$$
(9)$$\mathrm{Kappa}=\frac{\rho_{o}-\rho_{e}}{1-\rho_{e}}$$
(10)$$\mathrm{MCC}=\frac{\mathrm{TP}\times \mathrm{TN}-\mathrm{FP}\times \mathrm{FN}}{\sqrt{\left(\mathrm{TP}+\mathrm{FP}\right)\left(\mathrm{TP}+\mathrm{FN}\right)\left(\mathrm{TN}+\mathrm{FP}\right)\left(\mathrm{TN}+\mathrm{FN}\right)}}$$
(11)$$\mathrm{AUC}\left(\text{for single point}\right)=\frac{\left(\frac{\mathrm{TP}}{\mathrm{TP}+\mathrm{FN}}\right)+\left(\frac{\mathrm{TN}}{\mathrm{TN}+\mathrm{FP}}\right)}{2}$$
where FP and TP denote the false positive and true positive, while FN and TN represent the false negative and true negative, respectively; $\rho_{o}$ is the observed accuracy; and $\rho_{e}$ is the expected accuracy (random chance).

#### 3.3.2. NGBoost Classification Algorithm: Theoretical Background

NGBoost is a supervised natural gradient descent (NGD)-based boosting algorithm. It can be utilized for both probabilistic regression and classification tasks. In this study, the NGBoost-algorithm-based classification approach was explored for model training and evaluation purposes. A conceptual representation of the NGBoost model is shown in Figure 11.

In Figure 11, $x_{in}$ represents the given input features, *M* is the base learner, *T* is the prediction target, and θ represents the parameters of the target distribution. The NGBoost model generates a conditional probability distribution function $P_{\theta }\left(output\left(T\right)|input\left(x_{in}\right)\right)$ of each predicted outcome in the range of 0 to 1. The higher the value of the mentioned function, the higher the probability of predicting the data class accurately will be, and vice versa. The proposed framework used boosting to build a series of decision trees (DTs) with reduced loss during model training. In other words, each DT learned from the previous tree and improved the next tree to enhance the model’s performance. The hyperparameter’s values for the NGBoost algorithm used in the proposed method are given in Table 2. Detailed information on the NGBoost algorithm, along with source code implementation, is referenced here: https://stanfordmlgroup.github.io/projects/ngboost (accessed on 18 October 2021).

## 4. Performance Evaluation of Proposed Classifier

In this section, the performance of the proposed classifier against the performance metrics mentioned in Section 3.3.1 is evaluated and discussed thoroughly. In order to train a ML model for the current theft-detection problem based on the NGBoost classifier, the data framed at Stage-2 was fetched. To avoid the overfitting problem in the developed model, two techniques were adopted: first, by generating enough samples of both data classes using the MWMOTE algorithm; and second, by setting the values of the hyperparameters in such way that model did not overfit the training data. A 10-fold cross-validation (CV) method was employed to assess the performance of the proposed model, the corresponding outcomes of which are depicted in Table 3.

It can be observed in Table 3 and Figure 12 that the proposed model attained a very high accuracy and recall; i.e., 0.93 and 0.91, respectively. It is worthwhile to mention that accuracy and recall are two of the most frequently utilized performance evaluation metrics in most theft-detection frameworks. However, these metrics should not be viewed as a decisive measure for the final selection of a theft-detection model. For example, a higher accuracy value can be obtained in the presence of an imbalanced data class distribution. This is because the developed model in the mentioned scenario was highly oriented towards rightly classified majority class samples only. Similarly, the recall value lacked information regarding the number of false-positive cases, which consequently generated misleading outcomes and may have caused the commencement of false inspections due to wrongly classified theft consumers. Therefore, the efficacy of the proposed framework was further validated by measuring precision, Kappa, F1_score_, and MCC. The precision metric estimated the ability of the classifier to accurately classify the theft cases. The proposed theft-detection model achieved a high average precision value of 0.95. Another important performance evaluation metric is F1_score._ It is the harmonic mean between precision and recall. The F1_score_ for the current case was calculated as 0.92, which was significantly higher for any supervised ML classifier. Likewise, the MCC metric yielded a high score only if the prediction had achieved good scores in all four confusion matrix categories. The proposed classifier attained an average MCC value of 0.88.

Another important metric for evaluating the theft-detection models is the AUC [16,46]. The AUC facilitates extracting the information contained in a ROC curve. It illustrates the trade-off between the true positive rate (TPR) and false positive rate (FPR). In other words, it provides a direct measure of the classifier’s ability to correctly segregate the positive (theft) and negative (healthy) classes. In the current study, the proposed classifier attained an average AUC value of 0.96, as shown in Figure 13, which further signified the efficacy of the developed model.

In addition, the cumulative gain curve obtained using the proposed model is shown in Figure 14. The cumulative gain curve was used to visually understand the overall probabilistic predictions produced for each outcome by the classifier. The proposed model assigned each model outcome a probability score so that the higher the probabilistic prediction value, the higher the confidence in data class prediction would be, and vice versa. For each fraudster consumer predicted by a model, the predictions with the highest confidence scores should be addressed first for onsite inspection, and those with lower confidence scores may be discarded for a site inspection. In this way, human resource allocation for conducting site inspections can be reduced, and the theft detection hit rate can also be improved; whereas the threshold for setting the confidence value of model prediction can differ based on the distribution company’s resources.

The *x*-axis in the curve represents the percentage-wise proportion of consumers while the *y*-axis signifies the gain for all the model responses in a given data class. The class-0 and class-1 represent “Healthy” and “Theft” consumers, respectively. It can be observed in Figure 14 that the developed model outcome probability was 0.4 for 20% of the total consumers, while its value was greater than 0.8 for more than 45% of consumers.

Considering the outcomes of all the performance evaluation metrics, we concluded that the proposed theft-detection model correctly classified the majority of the Theft and Healthy cases in the provided dataset, and thus proved its essence and significance.

### 4.1. Confusion Matrix of the Proposed Model

The confusion matrix (CM) is a popular metric for assessing the performance of supervised classification models. It is applicable to both binary and multiclass classifications issues. It is essentially a square matrix, with the rows representing the actual class of the instance and the columns representing their predicted class, as illustrated in Figure 15.

Figure 16 illustrates the confusion matrix attained by the proposed model. Here, “0” represents the negative class, while “1” represents the positive class. For readability purposes, CM values were normalized in percentage form. It is evident in the above figure that the classifier correctly classified most of the Theft and Healthy consumers.

### 4.2. Outcomes Interpretability Using Tree SHAP Algorithm

In this section, the proposed theft-detection model’s model outcomes are explained by employing the tree SHAP algorithm developed by Lundberg et al. [37]. The aim of employing this algorithm was to assess each input feature’s contribution in predicting the model outcome. Figure 17 shows the top features that significantly contributed to the prediction of the target variable by using the tree SHAP algorithm. Furthermore, sensitivity analyses also were conducted that facilitated providing more insight into how input features contributed to the prediction of the model outcome [47]. In this case, the feature sensitivity of input features was computed by changing one or more features while keeping the rest constant. If the model’s predicted result varied significantly when a feature’s value was updated or dropped, then that feature had a major impact on prediction. The corresponding outcome attained by the sensitivity analysis technique for the proposed model is depicted in Figure 18.

It can be observed in Figure 17 and Figure 18 that the entropy feature (a measure of internal variations lying in given data) attained the highest value (in terms of both the tree SHAP algorithm and the sensitivity analysis technique) in predicting the target variable. To illustrate its significance, the kernel density estimation (KDE) plot of computed entropy values and a sample of a consumer’s consumption pattern relative to different entropy values are shown in Figure 19 and Figure 20, respectively. The KDE plot in Figure 19 shows that the fraudster consumers acquired a comparatively higher entropy than the healthy consumers. On the other hand, Figure 20 provides a Healthy and Theft consumer sample for two consecutive years (2014 and 2015). Unlike the Theft consumer, for whom the entropy was higher due to irregular consumption patterns during the similar months of the considered years, the Healthy consumer’s consumption patterns provided identical kWh curves and, consequently, lower entropy. Hence, by interpreting such in-depth information of selected features through the tree SHAP algorithm, numerous deciding factors involved in predicting the model’s outcomes could be effectively interpreted, which helped to enhance the performance of the developed model.

## 5. Results and Discussion

In this section, the efficacy of the proposed theft detection framework is assessed and compared with the latest gradient boosting decision trees and the most frequently used classical ML models under identical datasets. To accomplish the mentioned objective, the performance assessment of all the studied ML models was made based on five of the most widely utilized performance metrics (detailed in Section 3) using a 10-fold cross-validation technique. Since the proposed NGBoost algorithm was essentially a modified variant of an ensemble gradient descent decision tree, its performance was initially compared with similar types of models, such as the extra tree classifier, extreme gradient boosting classifier, CatBoost classifier, light gradient boosting machine, and Ada boost classifier. The outcomes of this comparative study are illustrated in Figure 21. Later, in order to show its efficacy against the classic ML methods, another comparative analysis was carried out using an identical dataset and similar performance evaluation metrics. The corresponding results are shown in Figure 22.

We concluded from the results depicted in Figure 20 and Figure 21 that the proposed electricity-theft-detection framework outperformed all the competing ML models in terms of seven of the most widely used performance evaluation metrics. It provides an accuracy of 93%, precision of 95%, and recall of 91%, which were significantly higher than all other considered classifiers.

## 6. Conclusions

In this study, a novel feature-engineered electric-theft-detection framework was developed using the intelligence of the NGBoost algorithm. For ease of understanding and presentation, the proposed framework was divided into three major stages: data preprocessing; feature engineering; and model training, testing, and interpretation. To effectively classify the consumers into healthy and fraudster categories, each stage was executed sequentially. Initially, the data preprocessing stage was commenced, in which the problem of missing entries in the acquired smart meter dataset was addressed by the random-forest-based imputation technique. Afterward, the feature engineering stage was initiated, in which the unbalancing of the target variable was tackled by the minority class oversampling method, called the MWMOTE algorithm. Later, the combination of TSFEL and the whale optimization algorithm was utilized to extract and select the most important features from the balanced dataset acquired at the end of the previous stage. In the final stage, the NGBoost algorithm was used for model training and testing purposes. To assess and validate the performance of the proposed theft-detection framework, its performance was compared with that of the latest gradient descent ML machines, ensemble boosting and bagging classifiers, and the classic ML models. The proposed framework achieved superior performance compared to the other models based on a few well-known performance assessment metrics such as accuracy, precision, recall, Kappa, and MCC values; thus, it validated its efficacy and essence.

## Figures and Tables

**Figure 1 sensors-21-08423-f001:**
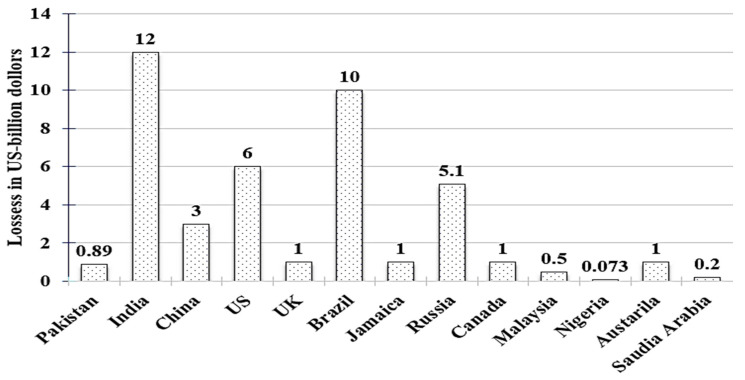
Nontechnical losses in different countries [3].

**Figure 2 sensors-21-08423-f002:**
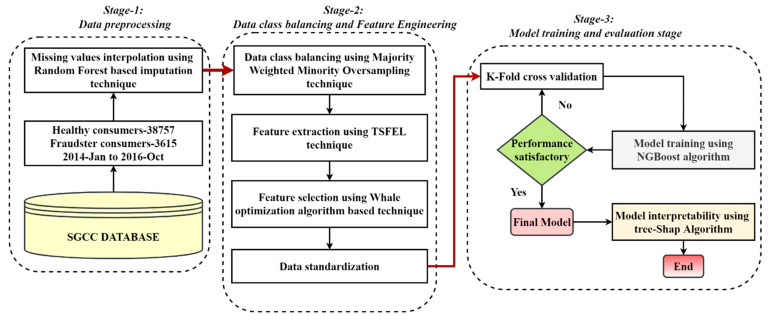
The simplified flow chart of the proposed feature-engineered–NGBoost classifier for electric theft detection.

**Figure 3 sensors-21-08423-f003:**
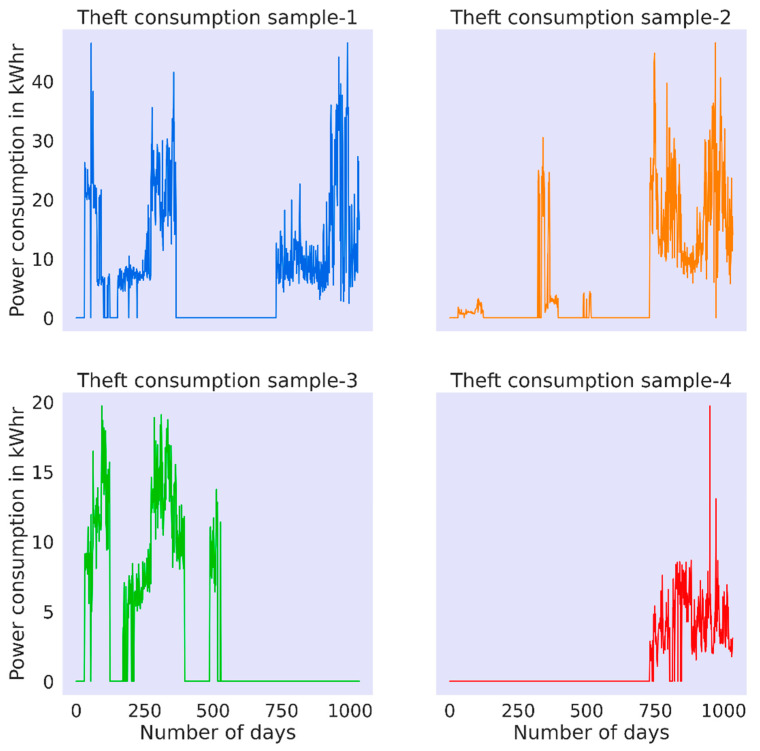
Fraudster consumers: electric consumption pattern in the acquired dataset.

**Figure 4 sensors-21-08423-f004:**
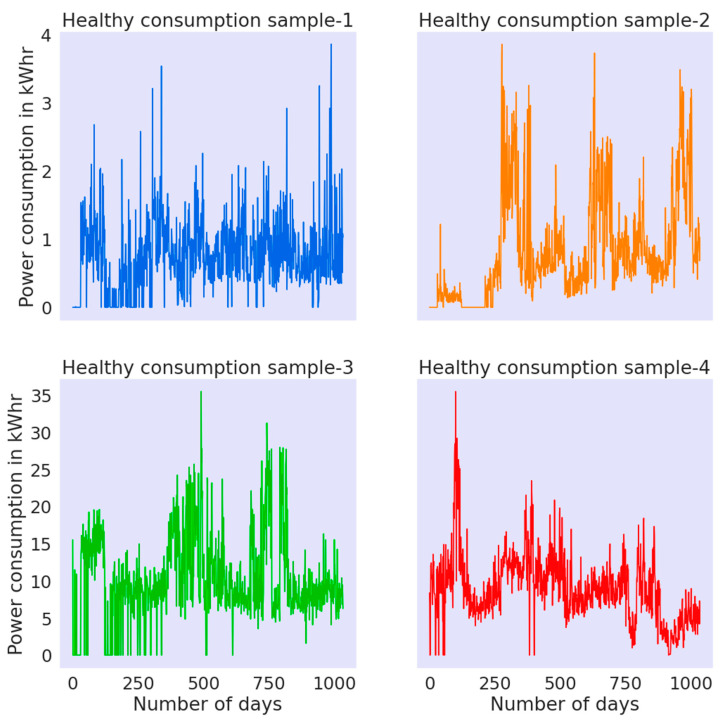
Healthy consumers: electric consumption pattern in the acquired dataset.

**Figure 5 sensors-21-08423-f005:**
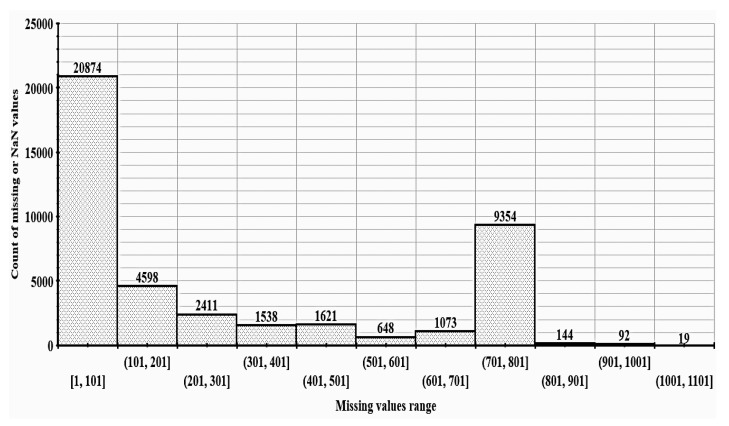
Histogram of missing values present in SGCC dataset.

**Figure 6 sensors-21-08423-f006:**
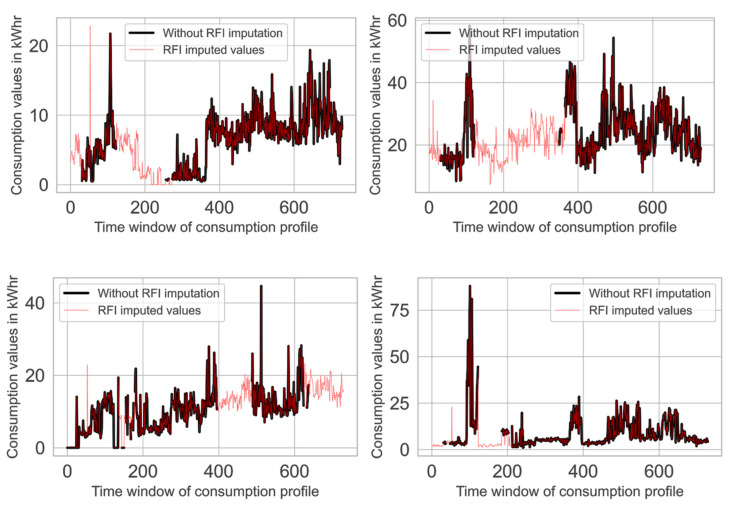
Missing value imputations using the random forest imputer.

**Figure 7 sensors-21-08423-f007:**
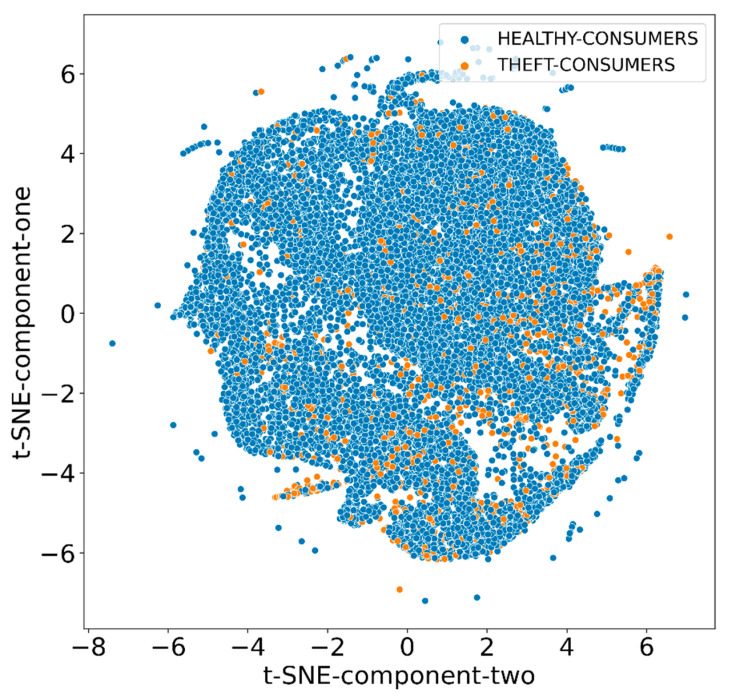
Data class distribution before using MWMOTE algorithm (imbalanced dataset).

**Figure 8 sensors-21-08423-f008:**
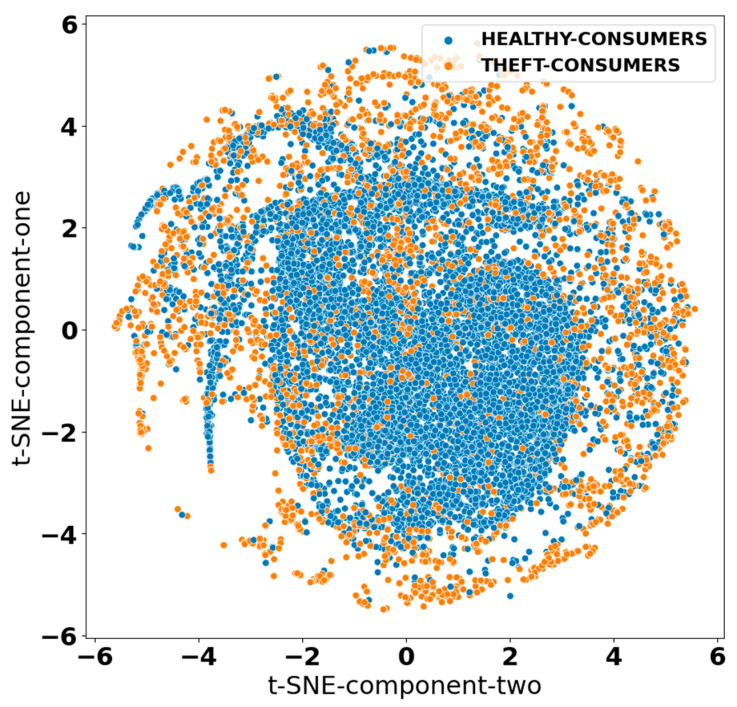
Data class distribution after using MWMOTE algorithm (balanced dataset).

**Figure 9 sensors-21-08423-f009:**
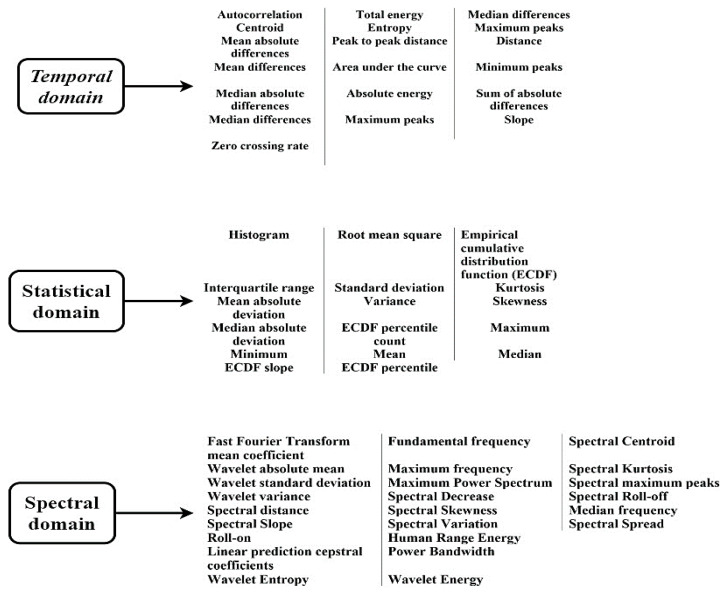
Extracted features using the TSFEL algorithm [37].

**Figure 10 sensors-21-08423-f010:**
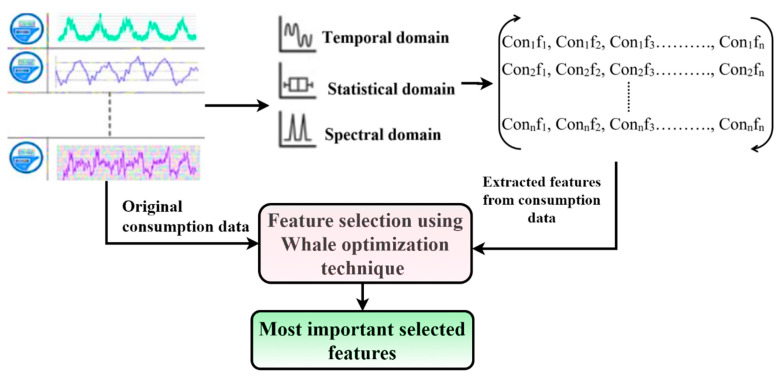
Feature selection using the WOA-FS technique.

**Figure 11 sensors-21-08423-f011:**
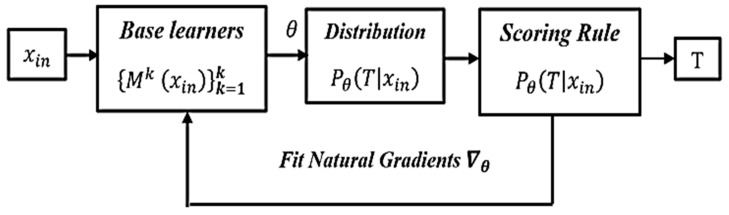
Block diagram representation of NGBoost model.

**Figure 12 sensors-21-08423-f012:**
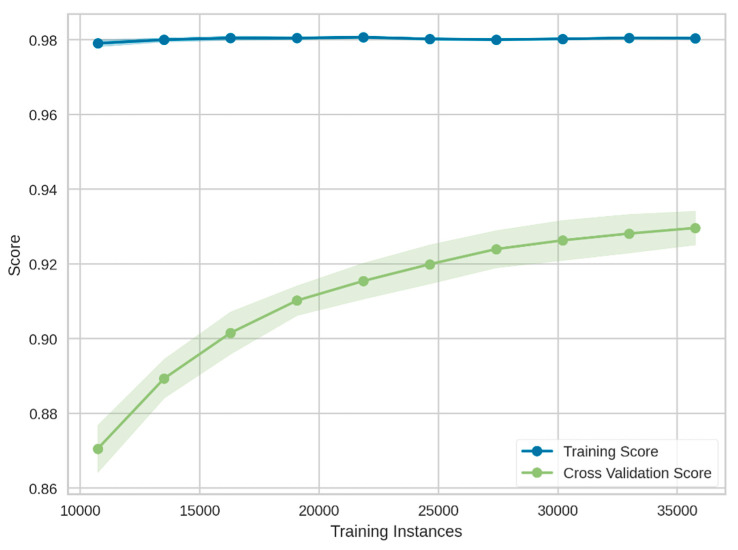
Training cross-validation accuracy score of the proposed model.

**Figure 13 sensors-21-08423-f013:**
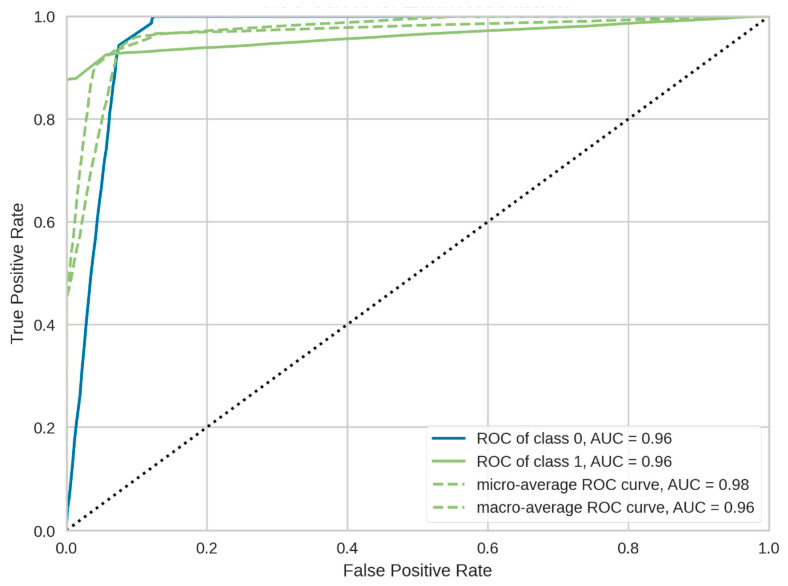
ROC curve of the feature-engineered–NGBoost classifier (proposed model).

**Figure 14 sensors-21-08423-f014:**
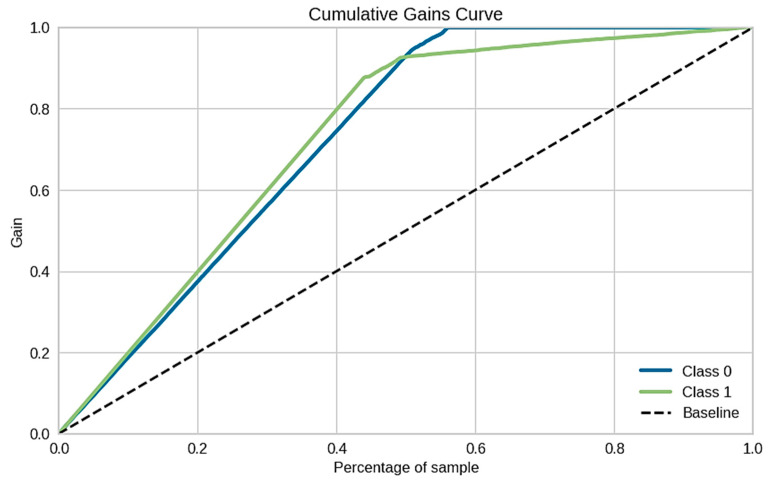
Cumulative gains curve of NGBoost classification algorithm.

**Figure 15 sensors-21-08423-f015:**
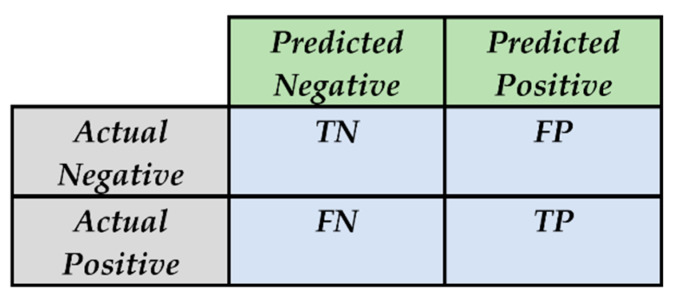
Confusion matrix for binary classification problem.

**Figure 16 sensors-21-08423-f016:**
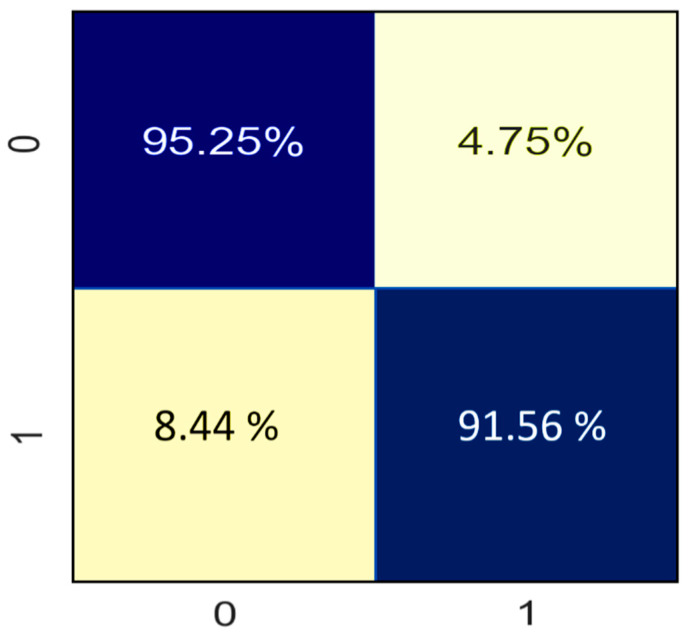
Confusion matrix of proposed theft-detection model.

**Figure 17 sensors-21-08423-f017:**
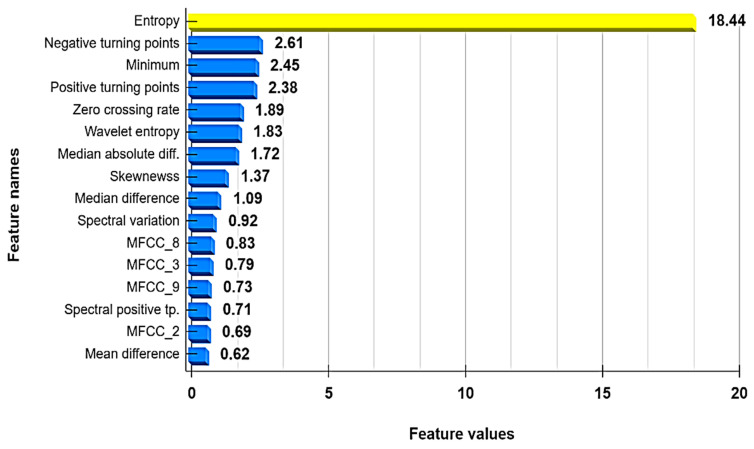
SHAP value of the proposed model.

**Figure 18 sensors-21-08423-f018:**
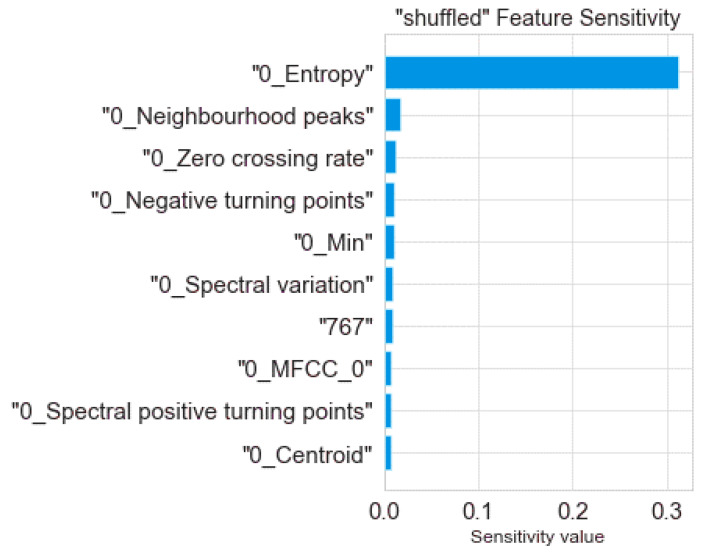
The sensitivity analysis of the proposed model.

**Figure 19 sensors-21-08423-f019:**
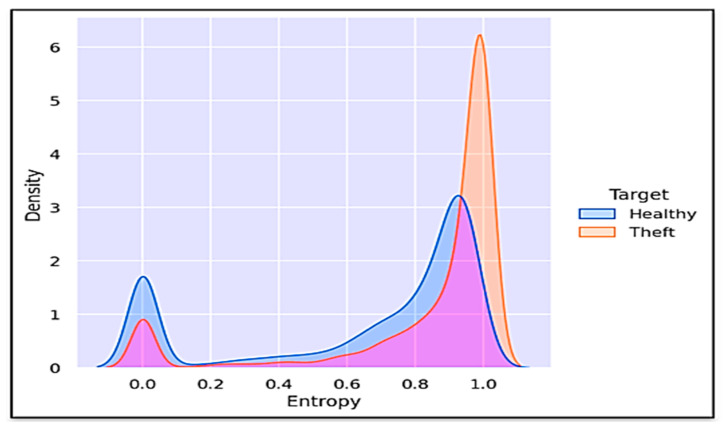
KDE plot of computed entropy values from consumers’ consumption data.

**Figure 20 sensors-21-08423-f020:**
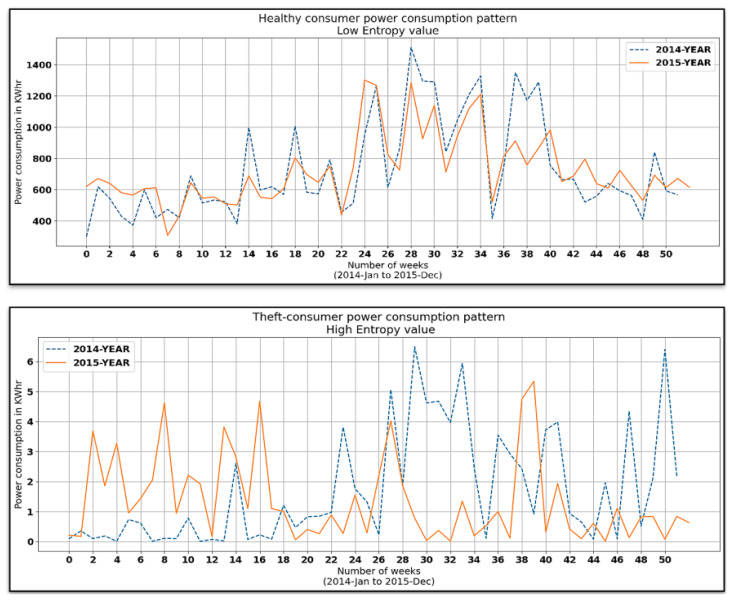
Consumers’ consumption patterns relative to different entropy values.

**Figure 21 sensors-21-08423-f021:**
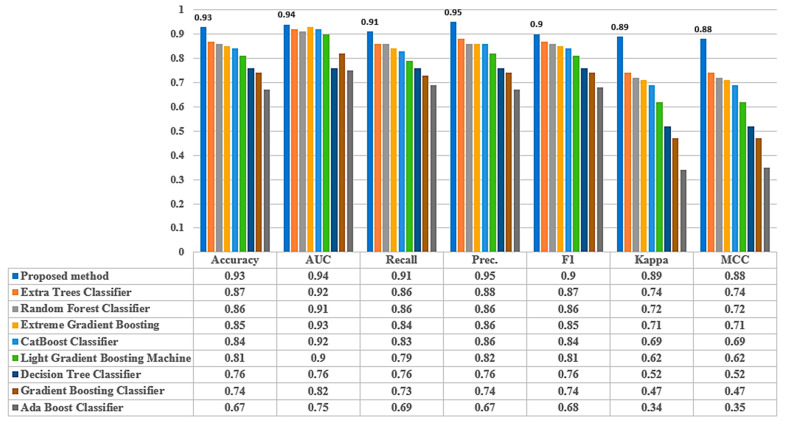
Performance evaluation of studied tree-based ML models.

**Figure 22 sensors-21-08423-f022:**
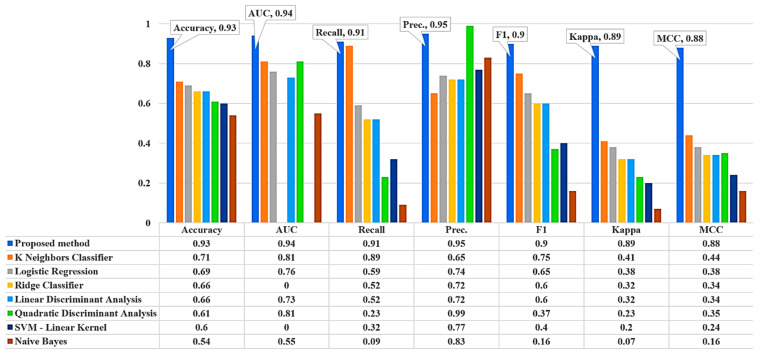
Performance evaluation of studied conventional ML models.

**Table 1 sensors-21-08423-t001:** Problems identified and the proposed solutions.

Problem Identified	Proposed Solution
Missing and inconsistent entries in data [9,10,11,13,14,15]	Supervised ML-based random forest imputation technique [29]
Data class imbalance [16,17,18,19,20]	Majority weighted minority oversampling technique algorithm [30]
Irrelevant and redundant features [31,32]	Time series and statistical-technique-based novel feature extraction using TSFEL algorithm [33]
High data dimensionality [26,34]	Feature selection using whale optimization algorithm [35]
Model selection [34,36,37]	Natural gradient boosting trees algorithm [38]
Model’s prediction interpretation	Tree SHAP additive explanations algorithm [39]
Reliable evaluation	AUC metric, precision, recall, Matthew’s correlation coefficient, Cohen’s kappa

**Table 2 sensors-21-08423-t002:** The hyperparameter’s values for the NGBoost algorithm used in the proposed method.

Parameter Name	Description	Parameter Value
learning_rate	Helps in setting weighting factors for the addition of new trees at each iteration to the classifier.	0.1
n_estimatiors	The number of boosting iterations to be performed.	100
subsample	The number of samples to be used for fitting the individual base learners. Optimal selection of this parameter can assist in setting bias and variance values.	0.5
min_samples_split	The minimum number of samples to be present at a leaf/internal node. This parameter controls the model overfitting/underfitting related problems.	5
min_samples_leaf	The minimum number of samples to be present at the leaf. Controlling this parameter helps in overfitting/underfitting-related issues.	6
max_depth	Helps in building the structure of the regression tree.	8
max_features	Number of features to be selected when searching for split.	15
max_leaf_nodes	Optimal selection of this value facilitats reducing the impurity of trees.	6
Tol	This value facilitates early stopping if there is no change in the loss.	0.20
Base_learner	Used to describe the base component of multiple classifier systems.	Regression trees
Probability_distribtuion	Normal distribution for continuous output, and Bernoulli for binary output.	Bernoulli
Scoring_rule	Maximum likelihood or continuous ranked probability score.	Maximum likelihood estimation

**Table 3 sensors-21-08423-t003:** The 10-fold cross-validation results for the proposed model.

*Performance Metric*	*Fold-1*	*Fold-2*	*Fold-3*	*Fold-4*	*Fold-5*	*Fold-6*	*Fold-7*	*Fold-8*	*Fold-9*	*Fold-10*	*Mean*
**Accuracy**	0.93	0.94	0.94	0.93	0.94	0.94	0.93	0.93	0.93	0.93	0.93
**Recall**	0.92	0.91	0.90	0.92	0.93	0.93	0.92	0.90	0.92	0.91	0.91
**Precision**	0.95	0.96	0.93	0.96	0.95	0.94	0.95	0.93	0.95	0.96	0.95
**Kappa**	0.86	0.88	0.89	0.95	0.88	0.88	0.86	0.87	0.9	0.9	0.89
**Fl_score_**	0.93	0.91	0.90	0.89	0.90	0.91	0.93	0.94	0.93	0.94	0.92
**AUC**	0.94	0.96	0.97	0.97	0.96	0.97	0.93	0.96	0.97	0.98	0.96
**MCC**	0.86	0.87	0.87	0.87	0.87	0.88	0.95	0.87	0.86	0.87	0.88

## Data Availability

The SGCC smart meter dataset used to support the findings of this study is cited in Section 3.1 line number 4 (38-reference), and it is publicly available on the following link https://github.com/henryRDlab/ElectricityTheftDetection, accessed on 19 October 2021.

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
