# Peer review of "A Novel Feature-Engineered–NGBoost Machine-Learning Framework for Fraud Detection in Electric Power Consumption Data"

_sensors, 2021, doi:10.3390/s21248423_

Round 1

Reviewer 1 Report

Dear Authors

This paper proposed a novel feature-engineered Natural gradient descent ensemble boosting (NGBoost) based electricity theft detection framework utilizing a supervised machine learning approach. The proposed framework is sequentially executed in three stages: data pre-processing, feature engineering, and model evaluation. The simulation results reveal that the proposed framework achieved an accuracy of 93%, recall of 91%, and precision of 95%.The paper deals with an interesting topic, although some minor and major observations must be addressed before a final decision can be taken.

  1. Please improve the quality of the pictures in the article, such as fig. 21, 22, etc.
  2. It is suggested to use "confusion matrix" to characterize and analyze the advantages of the proposed model.

Author Response

Reviewer: 1

Comments from Reviewer: This paper proposed a novel feature-engineered Natural gradient descent ensemble boosting (NGBoost) based electricity theft detection framework utilizing a supervised machine learning approach. The proposed framework is sequentially executed in three stages: data pre-processing, feature engineering, and model evaluation. The simulation results reveal that the proposed framework achieved an accuracy of 93%, recall of 91%, and precision of 95%.The paper deals with an interesting topic, although some minor and major observations must be addressed before a final decision can be taken.

Response to reviewer:

            Thank you for your kind comments and for summarizing our overall work so well. You have provided us with useful feedback which will help us further improve our work in the future.

Comments from Reviewer: Please improve the quality of the pictures in the article, such as fig. 21, 22, etc.

Response to reviewer:

            Thank you for highlighting. We have tried our best to enhance the quality of the mentioned figures, and all figures in general, in the revised version.

Comments from Reviewer: It is suggested to use "confusion matrix" to characterize and analyze the advantages of the proposed model.

Response to reviewer

             Thanks a lot for your excellent suggestion, in the revised version we have added the confusion matrix attained by the proposed model. Illustrated in Figure 15 and Figure 16 and mentioned in section 4.1.

Reviewer 2 Report

The paper deals with a supervised framework for electricity theft detection. The paper is clear and well-written in general. Regarding the novelty, it seems that the main contribution of the paper is the sequential application of some existing techniques to the problem at hand. It is not clear if the authors proposed some modifications to these techniques. The authors must also clarify some points detailed below:

  • The authors should make some brief comments on the related works cited in Section 2, highlighting their pros and cons;
  • Section 3.1:
    • Line 208: Which is the appropriate representation that the data is reshaped into?
    • Lines 212-213: The authors said the plots for consumption patterns for a few of the random samples from both fraudster and healthy consumers validate the genuineness of the acquired labeled data. How can this be possible? I think the word "validate" here is not the best one to be used in this sentence. These plots do really have some characteristics that can indicate the data is valid, but how can we prove that?
  • Section 3.2.1
    • The authors must cite references to MWMOTE and t-SNE;
    • How the identification of highly important minority class samples is done?
  • Section 3.2.2
    • The authors must cite references to TSFEL and WOA;
    • Line 291: Which supplementary information is that?
    • Line 308: The authors said the WOA algorithm can achieve the optimum solution. Is it possible to guarantee that the WOA algorithm will always achieve the global optimum for the problem dealt with in this paper? If this is not true, then the algorithm can achieve some (local) optimum solution;
    • The authors did not define the elements of the matrix appearing in Fig. 10;
    • Line 317: How are the features selected from the original data?
    • Eq. (1): Is there any relationship between $\alpha$ and $\beta$?
    • Line 327: Which termination criterion was used?
  • Section 3.3.2:
    • Fig. 12: Is the $\alpha$ used as the learning rate the same as that in (1)? Also, $I_S$, $fit$, and $\eta$ were not defined. In fact, the pseudo-code in this figure is the same as the one in reference [48].  Since the authors do not explain in details the NGBoost algorithm, nor propose any modifications to that, I think Fig. 12 does not add any information to this manuscript;
    • The authors must explain what (11) represents;
  • Section 4:
    • The authors must present the values of parameters and hyperparameters used in each algorithm used in the proposed framework;
    • Fig. 16: The importance values of the features showed in this figure are very low. In this case, why do the authors present this figure?
    • Fig. 18: In the vertical axes, which density is that?
    • Fig. 19: Why are the amplitudes of both plots so different?
  • Section 5: It is not clear if the other simulated algorithms, like Ada boost and lightGBM, just replace NGBoost in the proposed framework or if they directly use the raw data, without any pre-processing.
  • References 17 and 35 are incomplete (they misses the journals where they were published).

Author Response

Reviewer-2

  1. Comments by reviewer: The paper deals with a supervised framework for electricity theft detection. The paper is clear and well-written in general. Regarding the novelty, it seems that the main contribution of the paper is the sequential application of some existing techniques to the problem at hand. It is not clear if the authors proposed some modifications to these techniques.

Response by authors:

            Thanks a lot, respected reviewer, for your valuable comments. We gladly take these very important concerns regarding this research work. Indeed, the primary contribution of the work is to sequentially execute the most relevant and novel techniques in order to deal with the problem more effectively.

Following is a brief summary of the contribution of the proposed work

  • The theft detection model developed in this work is based on the “NGBoost classification algorithm”, which produces the classification outcomes based on probabilistic estimations. In contrast, the previously suggested theft detection frameworks (discussed in the literature review) do by point estimations. The problem with the point estimations type of predictions is that they produce a single outcome such as either “Theft” or “Healthy” for the given consumption profile. In contrast, the probabilistic type of predictions assigns each model’s prediction a confidence score from a scale of 0 to 1. This confidence score can assist in initiating on-site inspections so that predictions that attain a higher confidence score for positive outcomes, i.e., “theft,” must be addressed first, and any prediction with a lower confidence score can be omitted for a site inspection. In this way, human resource allocation can be reduced, whereas the threshold to set confidence value can vary depending on the distribution companies' available resources. The cumulative gain curve illustrated in Figure 15 which depicts overall probabilistic predictions produced for each outcome by the NGBoost classifier, and in the revised, we have added the explanation also.
  • In addition, for handling missing values in collected data, most previous work has either deleted completely or imputed missing values using simple statistical techniques such as average, median, or most frequent values. The complete removal causes loss of information and simple statistical techniques in the presence of outliers in the data, yield skewed and unrealistic results. In contrast to that, we have utilized the random forest algorithm that correctly predicts the missing values present in the consumption profile, thus enhancing the overall learning of the ML classifier.
  • The proposed MWMOTE technique efficiently balances the data class distribution, which further strengthens the developed model’s generalizing ability
  • TSFEL and WHALE optimization-based combination in the proposed work for feature engineering helps select the most relevant features, which further facilitates the classifier in comprehending complex and overlapping features.
  • The major contribution of this research work is the implementation of a novel sequentially executed theft detection framework to facilitate the power utilities in their campaign against the fraudster consumers. The proposed model achieves a low false-positive, high precision, and high detection rate; thus, it saves the cost, labor, effort, and time required for executing on-site inspections.

  1. Comments by reviewer: The authors should make some brief comments on the related works cited in Section 2, highlighting their pros and cons.

Response by authors:

            Thanks a lot, respected reviewer, for your comment. In table 1, the brief comments related to the problems identified in the literature and the proposed solutions have been made.

Section 3.1:

  1. Comments by reviewer: Line 208: Which is the appropriate representation that the data is reshaped into?

Response by authors:

            The original smart meter data contained multiple files of consumptions profiles and labeled information. We combined those and filtered out unnecessary information, and finally reshaped them in the following manner

Consumer: No

Day_1 consumption

Day_2

consumption

…………..

Day_1035

consumption

Target label

Cons_1

.

.

.

.

Theft

Cons_2

.

.

.

.

Theft

.

.

.

.

.

.

.

.

.

.

.

.

.

.

.

.

.

.

Cons_n

.

.

.

.

Healthy

  1. Comments by reviewer: Lines 212-213: The authors said the plots for consumption patterns for a few of the random samples from both fraudsters and healthy consumers validate the genuineness of the acquired labeled data. How can this be possible? I think the word "validate" here is not the best one to be used in this sentence. These plots do really have some characteristics that can indicate the data is valid, but how can we prove that?

Response by authors:

            Thanks for your valuable comment. We have removed the word “validate” from work in the revised version and replaced it with “explore”. We completely understand the raised concern by the respected reviewer as the data and labeled information to each consumer profile used in current work is taken from publicly available SGCC smart meter data[1]. (Link to download data:https://github.com/henryRDlab/ElectricityTheftDetection).

            Different authors cited the “SGCC data” several times, and they have validated their developed models against the mentioned data. Following that pattern, the authors have utilized the mentioned data in the current work. In addition to that, the limitation of information publicly available for a given dataset does not provide a further method to prove data authenticity, however upon exploring several consumption profiles randomly out of 43 thousand consumers of both consumer’s categories, we have observed that theft consumers profile does not contain periodicity and consistency in their consumption profile. In contrast to that, healthy consumption profiles are regular and follow some identical consumption throughout the year.

Section 3.2.1

  1. Comments by reviewer: The authors must cite references to MWMOTE and t-SNE;

Response by authors:

            Thanks for highlighting this; in the revised we have added the references.

  1. Comments by reviewer: How the identification of highly important minority class samples is done?

Response by authors:

            Thanks for your comment. In order to identify a set of highly important minority class samples to be used for generating the synthetic samples, the MWMOTE algorithm computes the nearest neighbors of each labeled minority class sample using the KNN algorithm. After that, samples containing mostly majority class nearest neighbors within their specified boundary are considered noisy and less critical, therefore omitted from synthetic samples generation. In contrast, those samples containing most of the minority class samples within their boundary are highly important. Finally, the average-linkage agglomerative clustering-based technique is employed to produce new samples from highly important categories of minority class samples.

Section 3.2.2

  1. Comments by reviewer: The authors must cite references to TSFEL and WOA;

Response by authors:

            Thanks for highlighting this. In the revised, we have added the references.

  1. Comments by reviewer: Line 291: Which supplementary information is that?

Response by authors:

            Thanks for your comment. The supplementary information refers to the number of features extracted from consumption data using the TSFEL technique. These features are based on the statistical, temporal, and spectral mathematical domains. All extracted features are depicted in Figure 10.

  1. Comments by reviewer: Line 308: The authors said the WOA algorithm could achieve the optimum solution. Is it possible to guarantee that the WOA algorithm will always achieve the global optimum for the problem dealt with in this paper? If this is not true, then the algorithm can achieve some (local) optimum solution.

Response by authors:

            Thanks for your valuable comments. The reviewer has rightly pointed out the problem of local optima stagnation of metaheuristic-based optimization algorithms. To avoid this issue and to ensure a global optimum solution for the studied optimization problem, the authors have simulated the model for 20 runs with identical optimization parameters. This article has quoted the most optimal solution among all the runs has been quoted here in this article.

  1. Comments by reviewer: The authors did not define the elements of the matrix appearing in Fig. 10;

Response by authors:

            Thanks for your comment. In the revised version, we have explained the matrix elements appearing in figure 10.

  1. Comments by reviewer: Line 317: How are the features selected from the original data?

Response by authors:

            Thanks for your comment. The actual consumption data and extracted features are fed to the WOA-feature selection technique. The WOA-feature selection technique, with the help of the fitness function defined, yields a number of most essential features set. A detailed explanation of features selections from original data is provided in lines 321-327.

  1. Comments by reviewer: Eq. (1): Is there any relationship between $\alpha$ and $\beta$?

Response by authors:

            Thanks for your comment.  In the revised version, we have explained both parameters. The  and  (1- ) manage the trade-off between the classification error rate to the number of selected features subset[2]. The parameter β is derived from the alpha, when alpha will decrease beta will increase.

  1. Comments by reviewer: Line 327: Which termination criterion was used?

Response by authors:

            Thank you for this query. Since the information regarding the global solution is completely ambiguous at the start of the optimization process, therefore iteration number has been chosen as the termination criteria in the current work.

Section 3.3.2:

  1. Comments by reviewer: Fig. 12: Is the $\alpha$ used as the learning rate the same as that in (1)? Also, $I_S$, $fit$, and $\eta$ were not defined. In fact, the pseudo-code in this figure is the same as the one in reference [48].  Since the authors do not explain in details the NGBoost algorithm, nor propose any modifications to that, I think Fig. 12 does not add any information to this manuscript;

Response by authors:

            Thanks for your comments. You are a correct respected reviewer; we have not proposed any modification in the NGBoost algorithm, that’s why in the revised version, we have removed figure 12 and equation 11. For interested readers, we have provided the Ngboost algorithm’s authors website where the detailed documentation and source code for algorithm implementation can be found .(https://stanfordmlgroup.github.io/projects/ngboost/).

  1. Comments by reviewer: The authors must explain what (11) represents;

Response by authors:

            Thanks for your comments. In the revised version, we have included the reference where the mathematical explanation and detailed documentation of the NGBoost algorithm can be found.

Section-4:

  1. Comments by reviewer: The authors must present the values of parameters and hyperparameters used in each algorithm used in the proposed framework

Response by authors:

            Thanks for your comment. In the revised version, we have provided the proposed algorithm hyperparameters' values, and their explanation in Table 2, The algorithms compared against the proposed technique are fetched from the scikit learn library[3], where the hyperparameter setting for them is made using the grid search technique. Since there are 15 different algorithms (decision trees, boosting and bagging, kernel, nearest neighbor, and regression base) used for comparison against the proposed technique, each algorithm has several parameters, and due to space limitation and a broader scope of the paper, details of the all the compared algorithm are not provided in the paper excluding the proposed technique.

  1. Comments by reviewer: Fig. 16: The importance values of the features showed in this figure are very low. In this case, why do the authors present this figure?

Response by authors:

            Thanks for your valuable comment. You are correct that the importance score attained by the features shown in the quoted figure is low compared to other important features, but since the features depicted in the figure are “Month names”, authors have deliberately shown this figure to emphasize the significance of each month in positively predicting a positive outcome.

  1. Comments by reviewer: Fig. 18: In the vertical axes, which density is that?

Response by authors:

            Thanks for your comment. The vertical axis depicts the density of entropy values of individual consumption profiles. It is computed using the Kernel Density Estimation (KDE) function. A KDE  function plot is a method for visualizing the distribution of observations in a dataset, and it is analogous to a histogram. The KDE plot represents the data using a continuous probability density curve in one or more dimensions.

  1. Comments by reviewer: Fig. 19: Why are the amplitudes of both plots so different?

Response by authors:

            Thanks for your comments. The mentioned plots are consumption profiles of theft and healthy consumers. The theft consumption profile illustrated in the figure for that particular case has lower consumption, whereas healthy consumer has high consumption, that’s why two different amplitudes. In both profiles, the y-axis depicts the power consumption and the x-axis the time window of consumption.

Section 5:

  1. Comments by reviewer: Section 5: It is not clear if the other simulated algorithms, like Ada boost and lightGBM, just replace NGBoost in the proposed framework or if they directly use the raw data without any pre-processing

Response by authors:

            Thanks for your comments. In order to compare classifiers fairly, all algorithms used for validation had identical data for training and testing. This data was similar to that provided to NGBoost (the proposed technique) during its analysis.

  1. Comments by reviewer: References 17 and 35 are incomplete (they misses the journals where they were published).

Response by authors:

Thanks for highlighting this.

Reference 17:

              R.-S. Jeng et al., "Missing data handling for meter data management system," in e-Energy '13:   Proceedings of the fourth international conference on Future energy systems, Berkeley California         USA 2013, pp. 275-276.

Reference 35:

  1. A. M. Pereira et al., "Multilayer perceptron neural networks training through charged system search and its application for non-technical losses detection," in IEEE PES Conference on   Innovative Smart Grid Technologies, Innovative Smart Grid Technologies Latin America, 2013:     IEEE, pp. 1-6 %@ 1467352748

In the revised version, we have corrected the references.

References for the reviewer:

[1]         Z. Zheng, Y. Yang, X. Niu, H.-N. Dai, and Y. Zhou, "Wide and Deep Convolutional Neural Networks for Electricity-Theft Detection to Secure Smart Grids," IEEE Transactions on Industrial Informatics, vol. 14, no. 4, pp. 1606-1615, 2018, doi: 10.1109/tii.2017.2785963.

[2]         M. Sharawi, H. M. Zawbaa, and E. Emary, "Feature selection approach based on whale optimization algorithm," 2017: IEEE, pp. 163-168 %@ 1509047263.

[3]         F. Pedregosa et al., "Scikit-learn: Machine learning in Python," the Journal of machine Learning research, vol. 12, pp. 2825-2830 %@ 1532-4435, 2011.

Reviewer 3 Report

This paper presents a supervised machine learning based feature engineered-NGBoost classifier for electricity theft detection. In general, this paper is well written and the topic is interesting. Here, there are some concerns of this reviewer:

1 How scalable is the proposed electricity theft detection approach?

2 The proposed method might be sensitive to the values of its main controlling parameter. How did you tune the parameters? Please elaborate on that.

3 What are the limitations of the presented approach in practical applications?

4 Please improve the quality of figures completely to improve the readability of this paper.

5 In this study, how to handle the problem of premature convergence in the used whale optimization algorithm? Generally speaking, one needs to monitor the population diversity during iteration and add perturbations to maintain the diversity when the diversity is insufficient. Kindly see the work presented in [A] for instance.

[A] Zhang Y, et al. Optimized extreme learning machine for power system transient stability prediction using synchrophasors. Mathematical Problems in Engineering. 2015 Nov 10; 2015.

6 As stated in this study, ensemble learning and feature engineering can significantly improve the performance of supervised machine learning techniques. Authors are recommended to include and review the previous work to highlight this point.

Author Response

Reviewer-3

Comments by reviewer: This paper presents a supervised machine learning based feature engineered-NGBoost classifier for electricity theft detection. In general, this paper is well written and the topic is interesting.

Response by authors:

            Thanks a lot, respected reviewer, for your encouraging comments and for summarizing our overall work. These comments will allow us to continue to do and improve our work in the future.

Here, there are some concerns of this reviewer:

  1. Comments by reviewer: How scalable is the proposed electricity theft detection approach?

Response by authors:

            Thanks a lot, respected reviewer. The proposed technique can be scaled to the large smart meter datasets and for a variety of scenarios. As the proposed technique is designed in such a way that it deals with all kinds of most frequently occurring problems in building supervised machine learning-based theft detection models; The problems which are addressed in the proposed work are 

  • Handling of missing values,
  • Data class imbalance
  • Relevant input features for model training,
  • Suitable selection of classifier, and
  • Model outcomes predictions.

            In addition to that design, the model for practical application suitability a  real-world smart meter annotated dataset (State grid corporation of China (SGCC)) of 43 thousand consumers with consumption data of 1035 days was used. The mentioned dataset is currently most cited and widely adopted for developing fraud detection models

            The smart dataset contains a lot of missing values; in order to tackle that in the proposed framework, RandomForest based missing values imputation technique has been proposed. Similarly, most publically available smart meter datasets contain very few fraudsters consumers, and that imbalance of data class’s distribution causes developed model’s lean towards the majority class to avoid in the proposed method effective data class distribution method based on MWMOTE algorithm was used.  TSFEL and WHALE optimization-based combination in the proposed work for feature engineering helps select the most relevant features, which further facilitates the classifier in comprehending complex and overlapping features.

  1. Comments by reviewer: The proposed method might be sensitive to the values of its main controlling parameter. How did you tune the parameters? Please elaborate on that.

Response by authors:

            Thanks a lot, respected reviewer. The proposed NGBoost algorithm, by default, select its most optimal parameters by using the Grid search method as available in Scikit learn library (https://scikit-learn.org/stable/modules/generated/sklearn.model_selection.GridSearchCV.html).

            In addition to that, in the revised version, Table 2 has been added, indicating the hyper-parameters and their values in the proposed method.

  1. Comments by reviewer: What are the limitations of the presented approach in practical applications?

Response by authors:

            Thanks for your comment. Currently, in practical application, most power distribution companies conduct manual site inspections to identify fraudster consumers; these approaches have low accuracy and are very time-consuming. In contrast to that, the proposed model can accurately identify fraudster consumers with an accuracy of 93% and a precision of 95%. However, the increased number of Fraudster and Healthy consumptions profiles for model training purposes can improve the developed model’s accuracy and hit rate, further to be more effective for practical application.

            In addition to that, the weather also plays a crucial role in power consumption; if detailed information such as temperature, humidity, and wind speed is included in the model training process, then the model may achieve higher accuracy, thus more suitability for practical application.

  1. Comments by reviewer: Please improve the quality of figures completely to improve the readability of this paper.

Response by authors:

            Thanks for highlighting this. In the revised version, we have tried our best efforts to improve the quality of the figures.

  1. Comments by reviewer: In this study, how to handle the problem of premature convergence in the used whale optimization algorithm? Generally speaking, one needs to monitor the population diversity during iteration and add perturbations to maintain the diversity when the diversity is insufficient. Kindly see the work presented in [A] for instance.

[A] Zhang Y, et al. Optimized extreme learning machine for power system transient stability prediction using synchrophasors. Mathematical Problems in Engineering. 2015 Nov 10; 2015.

Response by authors:

            Thanks for the suggested paper. We have gone through the suggested paper, and it is very good research work. We have also included the mentioned paper in our work in the revised version.

            You have rightly pointed out the problem of local optima stagnation due to the premature convergence of metaheuristic-based optimization algorithms that can be solved by retaining the population diversity. In the case of WOA, during the prey search phase, humpback whales randomly search for the prey according to the position of each other. The WOA uses random values greater than one or less than −1 to force the search agent to move far from a reference whale. As a result, instead of using the best search agent found so far, the WOA changes a search agent's position in the exploration phase using a randomly picked search agent. Hence, the WOA's capacity to search the solution space at random, even when it is approaching the optimal solution, allows it to retain population diversity and avoid premature convergence.

  1. Comments by reviewer: As stated in this study, ensemble learning and feature engineering can significantly improve the performance of supervised machine learning techniques. Authors are recommended to include and review the previous work to highlight this point.

Response by authors:

            Thanks for your valuable suggestions. We have reviewed and included previous work in the literature review section in the revised version.

Reviewer 4 Report

This system you presented doesn’t show any characteristics about its own strengths and weaknesses, the parameters it used to make its decision, why it reached and …. The ML model is explainable if we can understand how a specific node in a complex model technically influences the output and we can fundamentally understand how it arrived at a specific decision. This subject in more specifically, entails: Understanding the main tasks that affect the outcomes and explaining the decisions that are made by an algorithm. In where of this draft the response for these concerns can be found? You have worked on time series data, so why RF why not RNN???? Moreover, according to explanation in this paper, this is a hybrid model which in the field of AI modeling is not new.

  1. Title should be revised.
  2. English of the work suffers from syntax flaws, word repetition as well as long and vouge sentences. It should be associated with a major revision by native expert or proof editing systems.
  3. L22: approach??? Do you mean for ‘technique’???
  4. Abstract from the first line is vouge. You talk about a novel method or framework which reader cannot follow you.
  5. Keywords should be representative and available in Abstract. ‘supervised machine learning’, ‘classification’ are general terms and don’t correspond the specific of the work. using the name of machine learning in abbreviation often is not recommended for keyword. Strongly recommendation to be rechecked.
  6. L27: ‘class unbalancing’??? do you mean for ‘imbalanced data’????
  7. L35-36: the authors claimed that accuracy, recall and precision is greater than all the competing model, the questions is that what are these competing models? Then to have precision, recall, and accuracy you required multi class confusion matrix that can be defined depends on data and corresponding intervals while here you just used binary confusion matrix. This statement strongly is conflicting and must and have to be revised. Many papers such as https://link.springer.com/article/10.1007/s00366-021-01444-1; https://arxiv.org/abs/2008.05756 and ...
  8. Reference fig1?
  9. Whole of the organization should revised. You have Introduction and what is the reason for sub-header in Introduction? Introduction is a place to describe: What is the main problem? What are previous studies, their applied methods and their limitations or gap? What is going to be executed in your work? what gap of previous models is going to be filled? With what method and why? What motives for? What about the applicability and its feasibility? What is the main novelty and contribution of applied method?
  10. It is wondering that in sub-header of introduction before any literature discussion on available method the flowchart of applied technique has been given.
  11. From 1.1. several abbreviations without any description in first using can be found.
  12. In Abstract the authors say something else in compared to L111. These are different stories.
  13. The last paragraph of the introduction should be assigned to brief summary of applied method and bolded findings.
  14. Given results only uncovers the samples and won’t determine what variables have the most influence. Extraneous variables might interfere with the information and thus outcomes can be adversely impacted by the quality of the work. When you have different involved parameters definitely sensitivity analyses for model calibration must be carried out to show their influence on the results. Look at https://iwaponline.com/jh/article/22/3/562/72506/Updating-the-neural-network-sediment-load-models and ....
  15. Time series analysis cannot provide any generalization from a single study and is difficult in obtaining appropriate measures. It also suffers from accurately identifying the correct model to represent the data. Could you please highlight how these problems have been overcome?
  16. Sometimes, the past data of the time-series is not enough to predict the future. Multiple additional features should be taken into account to get good forecasts. How did you judge the efficiently dealing with outliers? How to efficiently deal with multiple periodicities? Your works severely suffers from responses to these questions.
  17. The literature review concerning the origin of NGBoost its applicability, its limitation and deficiencies are very weak. Must be updated look at https://arxiv.org/pdf/1910.03225.pdf. In a whole perspective the literature review is significantly weak and poor. Must be updated concerning the RFI (e.g. https://onlinelibrary.wiley.com/doi/10.1002/sam.11348, …), Whale optimization (in total view metaheuristic algorithms and their applicability in optimization) and … look at https://link.springer.com/article/10.1007/s00521-021-06544-z and ...
  18. Where is the information on how the overfitting problem has been overcome? With what technique? What are the internal adjusted characteristics? Where is the optimum model and corresponding topology?
  19. Whale optimization suffers from slow convergence speed, low accuracy, and easy to fall into local optimum. Where did you give solution for these shortcomings? With what criteria?
  20. Could you please highlight what the inputs are (In a table)? 

Round 2

Reviewer 3 Report

I would like to thank the authors for this revised version since most of my concerns have been addressed. However, there are still the following comments:

1 Regarding previous comment 5, the authors stated that “We have also included the mentioned paper in our work in the revised version”.  However, there is no the mentioned paper in the references of the revised manuscript. Please cite it to better explain how to handle the problem of premature convergence in this work.

2 Some figures are still not clear enough. Different figures, such as Figures 2 and 6, can be made clearer to improve the readability of this paper.

3 The literature review must be strengthened. Avoid lumping references as in [6-11], [12-16], and all others. Instead, summarize the main contribution of each referenced paper in a separate sentence.

Author Response

Reviewer-3

Comment 1:

I would like to thank the authors for this revised version since most of my concerns have been addressed.

Response to comment 1:

Thank you very much, respectable reviewer. We are delighted that our responses fulfilled your expectations. We are working very hard to respond to all of the comments to the best of our ability. We appreciate you taking the time to read article and provide feedback.

Comment 2:

Regarding previous comment 5, the authors stated that “We have also included the mentioned paper in our work in the revised version”.  However, there is no the mentioned paper in the references of the revised manuscript. Please cite it to better explain how to handle the problem of premature convergence in this work.

Response to comment 2:

We deeply regret for this inconvenience. In the revised version we have included the mentioned paper, section 3.3.2, line 338 to 343.

Comment-3:

Some figures are still not clear enough. Different figures, such as Figures 2 and 6, can be made clearer to improve the readability of this paper.

Response to comment 3:

Thanks for your valuable comments.

Figure-2 and 6 has been completely re-drawn in the revised version in order to enhance it’s quality.

Comment-4: The literature review must be strengthened. Avoid lumping references as in [6-11], [12-16], and all others. Instead, summarize the main contribution of each referenced paper in a separate sentence.

Response to reviewer:

Thanks for your comments. The lumped references such as [6-11] and [12-16] and others have been removed in revised version to include only most relevant papers.

Again, we really appreciate your comments on our article, and we have done our best to respond appropriately. Thanks

Reviewer 4 Report

The made effort for replying to the comments are warmly appreciated but in some cases the responses are totally deviated from the core of the question. As the template is using the line numbers the response for each comment should be assigned to corresponding number in the manuscript. Moreover, you have used more than 20 figures, which strongly recommended to reduce. You talked about the word limit but definitely the manuscript can be truncated, and trivial explanations can be removed. The important concerns in the responses are that they may be the question of future readers and thus without any doubt must be supported with extra descriptive texts in the manuscript. About the English, if you have a look at the text again several syntaxes can be found.

For example, #9, I asked how you did the multiclass confusion matrix while the formulation of given attributes is well recognized and is not the response for this comment. Read the comment again please. The response for #12 is not acceptable while in scientific paper first you have descriptions and then Figures or flowcharts or … in #13, I mean from sub-header 1.1 and in none place of the sentences I pointed to figures. #14 should be described in Abstract. In #16 definitely you should introduce the sensitivity and corresponding references because the base of sensitivity is not yours and you borrowed it in scientific border. Open window for the reader to know how they can calibrate the model using sensitivity analysis and what can be dedicated with such analysis. Read the #17 and see your response please. In #19, you should strictly explain in the text and guide the reader if they would like to know the possible improvements. #20, give the information in the text. #22, Please just let the reader know how you became assure about the accuracy of input data. did you do any uncertainty analysis?

Round 3

Reviewer 4 Report

Good Luck